# A Self-Adaptive 1D Convolutional Neural Network for Flight-State Identification

**DOI:** 10.3390/s19020275

**Published:** 2019-01-11

**Authors:** Xi Chen, Fotis Kopsaftopoulos, Qi Wu, He Ren, Fu-Kuo Chang

**Affiliations:** 1School of Civil Aviation, Nanjing University of Aeronautics and Astronautics, Nanjing 210016, China; renhe@comac.cc; 2Shanghai Engineering Research Center of Civil Aircraft Health Monitoring, Shanghai Aircraft Customer Service Co., Ltd., Shanghai 200241, China; 3Department of Mechanical, Aerospace, and Nuclear Engineering, Rensselaer Polytechnic Institute, Troy, NY 12180, USA; kopsaf@rpi.edu; 4School of Electronic, Information and Electrical Engineering, Shanghai Jiao Tong University, Shanghai 200240, China; wuqi7812@sjtu.edu.cn; 5Department of Aeronautics and Astronautics, Stanford University, Stanford, CA 94305, USA; fkchang@stanford.edu

**Keywords:** self-sensing wing, dual-tree complex-wavelet packet transformation, convolution neural network, grey-wolf optimizer, flight-state identification

## Abstract

The vibration of a wing structure in the air reflects coupled aerodynamic–mechanical responses under varying flight states that are defined by the angle of attack and airspeed. It is of great challenge to identify the flight state from the complex vibration signals. In this paper, a novel one-dimension convolutional neural network (CNN) is developed, which is able to automatically extract useful features from the structural vibration of a recently fabricated self-sensing wing through wind-tunnel experiments. The obtained signals are firstly decomposed into various subsignals with different frequency bands via dual-tree complex-wavelet packet transformation. Then, the reconstructed subsignals are selected to form the best combination for multichannel inputs of the CNN. A swarm-based evolutionary algorithm called grey-wolf optimizer is utilized to optimize a set of key parameters of the CNN, which saves considerable human efforts. Two case studies demonstrate the high identification accuracy and robustness of the proposed method over standard deep-learning methods in flight-state identification, thus providing new perspectives in self-awareness toward the next generation of intelligent air vehicles.

## 1. Introduction

A novel “fly-by-feel” (FBF) concept was recently proposed, inspired by the remarkable flying capabilities of birds in nature. This concept aims at the development of the next generation of intelligent air-vehicle structures that can “feel”, “think”, and “react” based on high-resolution state-sensing, awareness, and self-diagnostic capabilities [1,2]. This kind of bioinspired systems are able to sense the environment (e.g., temperature, pressure, and aerodynamic forces) at unprecedented length and time scales. In addition, it could think in real time and be aware of its current flight state and structural health condition. Further, such a system could react intelligently under various situations and achieve superior performance and agility with optimal flight control. Compared with current flying approaches, the FBF concept has the following advantages: (1) sensing the external environment and internal structural condition (stresses, strains, damage) using the same integrated intelligent structure [3], (2) being aware of the general flight state and some critical states (flutter, stall, etc.) through effective data interpretation from intelligent structure, (3) optimal decision-making and autonomous flight control based on self-awareness [2]. Toward this end, great challenges have been exerted on current structural design and data-processing techniques, with a departure from existing technologies.

Recent years have witnessed investigations into different sensing-network architectures and simulations [4,5,6,7]. A typical example is that a stretchable network made of polymer-based substrates was designed by the Structure and Composites Lab (SACL) at Stanford University. This network contains many micronodes with the ability to integrate microsensors, actuators, and electronics for multiple applications [8]. Based on the development of microfabrication techniques [9,10,11], a composite wing structure with the sensor network monolithically embedded was successfully fabricated [3], as shown in Figure 1. This multifunctional wing consists of four sensor networks, and each network is integrated with strain gauges, resistive temperature detectors (RTD), and piezoelectric lead–zirconate titanate (PZT) transducers. Specifically, the strain gauge measures wing-strain distribution and can used to identify the dangerous area. RTD detects the temperature distribution to provide temperature compensation [12]. PZT transducers can be used under both active and passive modes. In active mode, they are used for structural health monitoring for any damage detection, while in passive mode, wing structural vibrations during flying are measured to indicate the air dynamic characteristics [3].

The flight state is characterized by a set of critical variables, such as altitude, angle of attack, airspeed, and temperature, forming a flight-state vector that indicates a specific aerodynamic condition. For the self-sensing wing mentioned above, its flight-state vector is defined by a combination of airspeed and angle of attack, which can be reflected by the signals obtained from the embedded sensor network. Kopsaftopoulos and Chang proposed a stochastic global identification framework on the basis of a developed Vector-dependent Functionally Pooled (VFP) model [2,13,14]. It combines the state-space method and stochastic time-series analysis, and is able to capture and predict the structural dynamics and the aeroelastic response under multiple flight states. From another perspective, if some indicative features can be extracted from the continuous structural aerodynamic behavior, it is possible to identify the flight state directly using the limited feature information instead of detailed characterization of the structural responses. Therefore, effective feature extraction is crucial in flight-state identification from the sensing-signal space to the practical-state space.

Feature extraction is one of the key steps in data-analysis processes, which can be divided into two approaches, by manpower and by machine [15]. Features selected by human expertise are explainable and straightforward. For example, various statistical features can be calculated, such as the mean value, standard deviation, peak value, and kurtosis, from both time domain and frequency domain for time-series signals [16,17,18]. The authors have also proposed novel feature-extraction and selection methods to achieve highly important features with low colinearity in the previous research [19]. However, these approaches heavily rely on domain knowledge. There might be useful features that remain uncovered. Alternatively, converting raw data into a set of features can be complemented by automatic feature construction using deep learning, which belongs to the broad family of machine-learning algorithms using multiple layers for feature extraction and transformation [20]. The development of deep learning was boosted since 2006, when a multilayer feedforward neural network consisting of stacked restricted Boltzmann machines showed superior performance in data representation [21]. Many DL architectures for signal and information processing have been developed, including deep autoencoders (DAE) [22], deep belief networks (DBN) [23], convolutional neural networks (CNN) [24], recurrent neural networks (RNN) [25], and their modifications. Although various deep neural networks (DNN) can save considerable human efforts in feature extraction and achieve competitive results, there are certain number of parameters in the DNN structure that are difficult to adjust. To enable adaptive DNN configuration, many researchers have developed various metaheuristic methods for parameter optimization [26]. To list a few examples, in a modified DAE for vibrational signal analysis, an artificial fish-swarm algorithm was used to optimized the critical parameters of the DAE [27]. Classical algorithm particle-swarm optimization was employed for determination of the proper structure of a trained DBN and a CNN for rolling bearing-fault diagnosis [28,29]. Three popular algorithms, simulated annealing, differential evolution, and harmony search, were used to optimize a CNN on classifying the MNIST and CIFAR datasets [30]. The cuckoo search algorithm was implemented in conjunction with an RNN and two back propagation neural networks for fast convergence and local minima avoidance [31]. These optimized DNNs outperformed their original structures, and produced results comparable or even superior to human experts or classic machine-learning algorithms.

In this study, we intend to apply the deep-learning technique to address the flight-state identification of the self-sensing wing. The signals collected from the PZT sensors embedded in the wing structure reflect the coupled aerodynamic–mechanical responses under varying flight states, with each state characterized by a specific angle of attack (AoA) and airspeed, and kept constant during the data collection. Therefore, these noise-corrupted structural responses can represent different flight states, which can be treated as a multiclassification problem. The objective of this paper is the introduction and evaluation of a novel multilayer network method for accurate flight-state identification based on vibration signals. Specifically, a one-dimensional (1D) CNN structure is developed for automatic feature learning. Taking advantage of the multichannel property of the CNN, reconstructed subsignals at different frequency bands via dual-tree complex-wavelet packet analysis are obtained as network inputs. A gray-wolf optimizer (GWO) [32] is then employed to determine the CNN key parameters. The proposed method does not only have strong feature-learning capabilities, but can also self-adapt to signal features for better classification performance without human intervention. Finally, we compare the flight-state identification accuracy with the standard CNN and other machine-learning models, and visualize the hierarchical feature-learning process through t-distributed stochastic neighbor embedding (t-SNE) [33]. The framework from the data acquisition, methodology development, evaluation, and application is shown in Figure 2.

The rest of paper is organized as follows: Section 2 presents the problem statement of this study. Section 3 develops the adaptive multichannel 1D CNN method using decomposed signals with parameter optimization by GWO. Two case studies with regard to general flight-state identification and stall detection and alerting are presented in Section 4, followed by their results and discussions in Section 5. Concluding marks are finally made.

## 2. Problem Statement

The problem statement of this work is as follows. Based on the sensing signals collected from a series of wind-tunnel experiments under varying flight states, we aimed to make the self-sensing wing accurately and automatically identify its undergoing flight state from the vibration time series. This problem can be further divided into three successive subproblems: (1) whether useful information can be learnt in an automatic manner, which are good at distinguishing different flight states; (2) how to establish the mapping relationship from the signal space to the physical flying state space; (3) how to improve the identification accuracy and robustness. 

Accordingly, the first two problems are addressed via the development of a 1D CNN that is capable of autonomous feature learning in a layer-wise approach. The mapping relationship is established after a certain number of network operations, in which useful features are obtained as data representation of the original raw data to indicate different states. The third problem is tackled through two approaches. The first approach is to decompose the preprocessed signals into various signal segments using dual-tree complex-wavelet packet transformation (DTCWPT) and the reconstructed signals at different levels with different combinations are used as multichannel inputs. The second approach is to optimize the CNN hyperparameters by applying a GWO. The above developed method is evaluated for multiple flight-state identification and then applied to the specific case of stall detection and alerting.

The main novel aspects of this study include:
(1)A tailored 1D deep CNN structure with multiple input channels using DTCWPT is developed for automatic feature learning instead of feature extraction and selection by human experts.(2)A self-adaptive CNN is proposed by combining the 1D CNN with a swarm-based GWO for automatic parameter determination instead of relying on human experience.(3)The flight-state identification of the self-sensing wing is treated as a classification problem by directly establishing the mapping relationship from the raw data to the physical space characterized by varying angle of attack and airspeed through wind tunnel experiments.(4)The application on stall detection and alerting with high identification accuracy provides new perspectives for autonomous flight control towards the “fly-by-feel” air vehicles.

## 3. Methodology Development

In this section, a self-adaptive 1D CNN method is proposed for flight-state identification of the self-sensing wing including three parts. Firstly, the basic theory of CNN is introduced. Then, a 1D deep CNN structure with DTCWPT is developed for autonomous feature extraction from the reconstructed subsignals. Lastly, a GWO is used to optimize CNN hyperparameters. The realization of the methods is programmed using the Python language (v 3.6.2) with tensorflow 1.8. 

### 3.1. Basic CNN Theory 

CNN is one of the active models in deep learning and has been widely applied in fields such as computer vision [34], speech recognition [35], and fault diagnosis [24]. Different from a traditional neural network with full connection throughout each layer, a CNN significantly reduces the network parameters by local connectivity and weights sharing using convolutional layers. This core building block consists of a set of kernels (or filters) which have a small receptive field. Each kernel moves across the input volume in a specified manner performing the convolution operation. Meanwhile, the kernel parameters remain the same to control the total number of free parameters. For a convolutional layer in the *l*th layer, the computation is expressed as
(1)$$X_{k}^{\left(l\right)}=f(\sum_{c}W_{k}^{(l),c}*X^{(l-1),c}+B_{k}^{l})$$
where *k* denotes the kernel number, *c* represents the channel number of the input $X^{\left(l-1\right)}$. $W_{k}^{(l),c}$ is the *k*th convolutional kernel corresponding to the *c*th channel, and $B_{k}^{l}$ is the learnable bias corresponding to the *k*th kernel, f(·) is the activation function and ∗ is the elementwise multiplication [28].

Another import building block is the pooling layer, which is a form of nonlinear subsampling. A pooling layer is commonly inserted between successive convolutional layers for parameters reduction with the intuition that the rough location relative to other features are more important than the exact location. Common subsampling functions are max pooling by selecting the maximum value in each kernel and mean pooling by averaging each kernel. After a number of convolutional and pooling layers, fully connected layers are attached for high-level reasoning, which are identical to the layers in the classic neural networks. 

The CNN training mechanism is based on backpropagation. Between densely connected layers, an error term δ of the *l*th layer is defined as
(2)$$\delta^{\left(l\right)}=({\left(W^{\left(l\right)}\right)}^{T}\delta^{\left(l+1\right)})\cdot f\prime (z^{\left(l\right)})$$
where $W^{\left(l\right)}$ is the parameter matrix of the *l*th layer, $z^{\left(l\right)}$ denotes the total weighted sum of inputs in layer *l* including the bias term, e.g., $z^{\left(l\right)}=W^{\left(l-1\right)}a^{\left(l-1\right)}+b^{\left(l-1\right)}$, $a^{\left(l-1\right)}=f(z^{\left(l-1\right)})$, which is the activation value in layer *l-1*. For output layer $n_{l}$, $\delta^{\left(n_{l}\right)}=\frac{\partial J}{\partial z^{\left(n_{l}\right)}}$, where *J* stands for the cost function, f′(·) is the derivative of the activation function.

For pooling layers, the error term of the *l*th layer is computed as
(3)$$\delta_{k}^{\left(l\right)}=u({\left(W_{k}^{\left(l\right)}\right)}^{T}\delta_{k}^{\left(l+1\right)})\cdot f\prime (z_{k}^{\left(l\right)})$$
where *k* is the kernel number as Equation (1) and $W_{k}^{l}$ is the *k*th kernel, u(·) is the upsample operation which propagates the error through the pooling layer by calculating the error in regard of the incoming to the pooling layer. 

Finally, for convolutional layers, the error term of *l*th layer is propagated through as
(4)$$\delta_{k}^{\left(l\right)}=conv(\delta_{k}^{\left(l+1\right)},\mathrm{rot}180(W_{k}^{l+1}))\cdot f\prime (z_{k}^{\left(l\right)})$$
where conv(·) is the convolution operation and rot180(·) denotes the 180 degrees rotation to make the convolution function perform cross-correction.

### 3.2. 1D CNN with DTCWPT

For signals with multimode aliasing, decomposition is necessary to achieve higher quality subsignals with different frequency bands, some of which may be better for classification. Compared with the conventional discrete wavelet packet transformation, DTCWPT provides a more precise frequency-band partition with approximate shift invariance [36]. It achieves signal decomposition and reconstruction through two parallel transforms of discrete wavelet, both of the which meet the reconstruction condition and each of the trees contains a set of low-pass and high-pass filters [37].

Herein, a sample signal is firstly decomposed and the subnodes at the same level or different levels are reconstructed to form the subsignals with the same length as the original signal. The reconstructed subsignals are then used as the CNN input instead of the original sample signal. To enable nonexperts’ intervention in 1D signal analysis for pattern recognition, a tailored deep CNN structure is developed as shown in Figure 3. Each vertical bar represents a network layer with different numbers of neurons corresponding to the exact data points. The multichannel inputs receive the reconstructed subsignals via DTCWPT. Features are extracted through several convolutional and maxpooling operation with increasing feature maps. A much shorter but deeper feature set is obtained and flatten to form a fully connected layer followed by a Softmax layer for multiclassification.

The configuration of the 1D deep CNN model used in this paper consists of an input layer, a convolutional layer C1, a pooling layer P1, a convolutional layer C2, a pooling layer P2, a convolutional layer C3, a pooling layer P3, a fully connected layer FC, and an output layer. For C1 to C3, the kernel numbers are 64, 256, and 128. The first kernel size is set to 10 while the other two sizes are denoted as *s*. For P1 to P3, max pooling is used, and the subregions are nonoverlapping with a size of 4. The learning rate is η, dropout is used in the last layer, and the dropout rate is λ. Relu is used as the activation function, and adam is used as the network training algorithm.

### 3.3. Parameter Optimization by GWO

In terms of the determination of the CNN key parameters, there is no mature method in theory. In this paper, we employ a swarm-based method called Grey Wolf Optimizer (GWO), which is one of the latest additions to the group of nature inspired optimization heuristics, to determine the optimum parameters of the trained 1D CNN. The GWO is inspired by the leadership hierarchy and hunting mechanism of grey wolves in nature and have demonstrated competitive results against some well-known evolutionary algorithms such as particle swarm optimization, genetic algorithm, and differential evolution [32]. The population of GWO is divided into four hierarchies. The first three fittest solutions are alpha (α), beta (β), and delta (δ), which guide other wolves omega (ω) for hunting. The main hunting phase include: encircling, hunting, attacking, and searching.

In encircling, the wolves update their positions according to the prey as follows:
(5)$$\vec{D}=\left|\vec{C}\cdot {\vec{X}}_{p}(t)-\vec{X}(t)\right|$$
(6)$$\vec{X}(t+1)={\vec{X}}_{p}(t)-\vec{A}\cdot \vec{D}$$
where *t* is the current iteration, ${\vec{X}}_{p}$ is the position vector of the prey, while $\vec{X}$ denotes the position vector of a grey wolf. $\vec{A}$ and $\vec{C}$ are coefficient vectors, $\vec{A}=2\vec{a}\cdot {\vec{r}}_{1}-\vec{a}$, $\vec{C}=2{\vec{r}}_{2}$, in which $\vec{a}$ is linearly decreased from 2 to 0 over the course of iterations, and ${\vec{r}}_{1}$, ${\vec{r}}_{2}$ are random vectors in [0,1].

During hunting, all wolves are obliged to update their positions according to first three best solutions obtained from encircling as follows:
(7)$${\vec{X}}_{1}={\vec{X}}_{\alpha }-{\vec{A}}_{1}\cdot {\vec{D}}_{\alpha }$$
(8)$${\vec{X}}_{2}={\vec{X}}_{\beta }-{\vec{A}}_{2}\cdot {\vec{D}}_{\beta }$$
(9)$${\vec{X}}_{3}={\vec{X}}_{\delta }-{\vec{A}}_{3}\cdot {\vec{D}}_{\delta }$$
(10)$$\vec{X}(t+1)=\frac{{\vec{X}}_{1}+{\vec{X}}_{2}+{\vec{X}}_{3}}{3}$$
where ${\vec{X}}_{\alpha }$, ${\vec{X}}_{\beta }$, and ${\vec{X}}_{\delta }$ are the positions of alpha, beta and delta. ${\vec{D}}_{\alpha }$, ${\vec{D}}_{\beta }$ and ${\vec{D}}_{\delta }$ are calculated using Equation (5) with different coefficient $\vec{C}$.

Attacking occurs when |A| < 1, otherwise, wolves diverge from each other for searching, which emphasizes further global exploration.

The optimization procedure of 1D deep CNN by GWO is illustrated in the following steps:

**Step 1:** Prepare the trained 1D deep CNN and set the complementary of the classification accuracy to be the fitness of GWO.

**Step 2:** Initialize the grey-wolf population $X_{i}=[x_{1i},x_{2i},\ldots x_{ni}]$, where *X* denotes the vector of parameters from CNN to be optimized.

**Step 3:** Initialize the GWO parameters *a*, *A,* and *C*.

**Step 4:** Train the 1D deep CNN with the initialized parameters corresponding to each agent.

**Step 5:** Evaluate the fitness of each agent and obtain the top three agents $X_{\alpha }$, $X_{\beta }$, and $X_{\delta }$.

**Step 6:** Update the position of each search agent by Equation (10).

**Step 7:** Update the GWO parameters *a*, *A,* and *C* and return to Step 5 to update the top three agents $X_{\alpha }$, $X_{\beta }$, and $X_{\delta }$ until the maximum iteration is reached.

**Step 8:** Return $X_{\alpha }$ as the optimized 1D deep CNN parameter vector.

## 4. Case Study

In this section, data collection and preparation from a series of wind-tunnel experiments are introduced. Then two cases are presented including a general flight-state identification with 16 classes for methodology demonstration and a novel application on stall detection and alerting with 12 classes.

### 4.1. Wind-Tunnel Experiment and Data Preparation

A series of wind-tunnel experiments of the self-sensing composite wing were conducted under various angles of attack (AoAs) and freestream velocities at Stanford University. The AoAs range from 0 degree up to 18 degrees with an incremental step of 1 degree. At each degree, data were collected for all velocities ranging from 9–22 m/s (incremental step of 1 m/s). For experimental details, please see Reference [2]. 

A series of wind-tunnel experiments of the self-sensing composite wing were conducted under various angles of attack (AoAs) and freestream velocities at Stanford University. The AoAs range from 0 degree up to 18 degrees with an incremental step of 1 degree. At each degree, data were collected for all velocities in the range of 9–22 m/s (incremental step of 1 m/s). For experimental details, please see Reference [2].

PZT signals reflect the coupled airflow-structural dynamics through the wing structural vibration and each time series contains coupled behavior with repeated patterns of a certain flight state. In each experiment, the structural vibration responses (90,000 data points) were recorded from the PZT located near the wing root at 1000 Hz sampling frequency. For each flight state, data are prepared in two steps: (1) the entire signal of 90,000 data points is divided into 180 segments (500 data points for each segment) to ensure enough samples for training while each segment has sufficient data points for feature extraction; (2) all signal samples are subtracted by their mean values eliminate the influence of zero drift. 

To evaluate the effectiveness of the proposed method and apply it for dangerous state prewarning, two sets of data are collected for general flight-state identification and stall detection and alerting.

### 4.2. General Flight-State Identification

The first dataset includes PZT signals covering a relative coarse resolution of 16 flight states corresponding to a combination of four AoAs (3, 5, 7, 9 degrees) and four airspeeds (12, 14, 16, 18 m/s). An example of signal segments representing 16 flight states is shown in Figure 4. The vertical axes are the signal amplitudes and the horizontal axes are the signal lengths.

It can be observed from Figure 4 that the amplitude of the vibration signal tends to be larger with the increasing AoAs and the airspeeds. However, most of the adjacent flight states can hardly be distinguished without detailed investigation.

### 4.3. Stall Detection and Alerting

The second dataset covers a dense resolution of flight states (AoAs: 11, 12, 13 degrees, airspeeds: 10, 11, 12, 13 m/s) for critical-state alerting. In aerodynamics, stall phenomenon is one of the dangerous conditions wherein a sudden reduction of the lift coefficient occurs as the angle of attack increases beyond a critical point. According to a previous analysis [2], the signal energy can be used as an indicator of the lift loss of the self-sensing wing. From the wind-tunnel experiments, the mean values of the signal energy for a series of AoAs (0–17 degrees) under four airspeeds (10, 11, 12, 13 m/s) are obtained and shown in Figure 5.

Signal-energy variation with respect to the angle of attack is similar under four different airspeeds. It is noticed that, for all velocities, energy starts to increase at approximately 12 degrees. For 10 m/s, a stall happens at 13 degrees with a large increase of energy, while for the other three velocities, a stall occurs one step late at 14 degrees. To be conservative, we defined the orange-shaded area starting from 13 degrees as the Stall region, which should be avoided. The initial increase region around 12 degrees is defined as the Alert region, in which warnings should be provided to the flight control for angle reduction when the self-sensing wing flies in this region. The left side before 12 degrees is the Safe region as shaded in light green.

## 5. Results and Discussion

### 5.1. General Flight-State Identification

The proposed method (DTCWPT-CNN-GWO) is evaluated through the general flight-state identification.

#### 5.1.1. Signal Decomposition and Reconstructed-Signal Selection

Firstly, a three-layer DTCWPT is employed to decompose the signals into various components with different frequency bands as shown in Figure 6. Reconstructions are then conducted for all three level signals without combination and obtained totally 14 subsignals from ‘a’ at the first level to ‘ddd’ at the bottom level as shown in Figure 6 and Figure 7. The vertical axes are the reconstructed signal amplitudes and the horizontal axes are the signal lengths.

The signal samples are randomly divided into 80% training set and 20% test set with equal classification category. The training set are further split into four subsets for cross-validation. The undetermined CNN parameters (the kernel size in the last two convolutional layers, the learning rate, and the dropout rate) [s,η,λ] are set as 10, 0.01, and 0.5. The objective of the primary classification experiment is to evaluate the classification ability of each reconstructed subsignals. The average identification accuracies using different subsignals as single input through the proposed CNN structure are shown in Figure 8.

It can be observed that the subsignals reconstructed from the first two decomposition levels have higher classification ability, among which the high-frequency band signal ‘d’ at the first decomposition level has the best performance. The top four subsignals are selected, ‘a’, ‘d’, ‘ad’, and ‘dd’, for the following analysis.

The second classification experiment is to make use of the multichannel mechanism of the CNN input layer and find the best combination from the four subsignals. The identification accuracies of various combinations are shown in Figure 9.

The last orange bar in Figure 9 indicates the classification accuracy using the original signal after preprocessing without DTCWPT. It is found that some subsignal combinations as multichannel inputs outperformance the original signal as single channel input, among which the three-channel combination ‘a/d/ad’ achieves the highest accuracy as 82%, improving more than 6% compared with the original signal.

#### 5.1.2. Parameter Optimization and Identification Accuracy Comparison

The third experiment was to employ GWO to optimize three key parameters, kernel size s, learning rate η, and dropout rate λ. The classification error was set as the fitness function. The CNN parameter setting with and without GWO is shown in Table 1.

To examine the effectiveness of the proposed method (DTCWPT-CNN-GWO), three other methods were used for comparison, a normal 1D deep CNN, a Deep Neural Network (DNN), and a classic Back-Propagation Network (BPN).

The structure and parameter setting of the other three methods are as follows: The determination of all parameters is by experience and repeated experiments.

The structure of DNN is 500-250-250-100-16, ‘relu’ was used as the activation function, ‘adam’ was used as the training algorithm, the learning rate is 0.001, and the maximum iteration is 100. 

The structure of BPN was 500-250-16, ‘relu’ was used as the activation function, ‘adam’ was used as the training algorithm, the learning rate was 0.001, and the maximum iteration was 200. The determination of all parameters is determined by experience and repeated experiments. 

The training and testing were conducted for 10 times; the identification accuracies of the four methods are shown in Figure 10 and the related information is summarized in Table 2.

The results show that: (1) the proposed method has the highest identification accuracy among the four methods, especially surpassing the standard 1D CNN by approximately four points. This proves that the proposed method can adaptively deal with aerodynamic–mechanical coupled structural vibration and achieve robust classification results. (2) Both CNN methods have much higher accuracies than the DNN and BPN. It can be explained that the local connection mechanism in CNN significantly improves the feature-learning performance compared to the conventional full connection networks, which have very limited classification ability using preprocessed signal data without feature extraction by manpower.

#### 5.1.3. Visualization of the Learning Process

To examine the feature-learning effects in the hierarchical CNN structure of the proposed method, 3D visualization is conducted through dimension reduction using a relatively new method named t-Distributed Stochastic Neighbor Embedding (t-SNE). It is a manifold learning technique by mapping to probability distributions through affine transformation, which is particularly suitable for nonlinear and high-dimensional datasets [33]. The learning results for five selected layers of the CNN are visualized sequentially in Figure 11. Sixteen colors represent 16 flight states, and a single point in each 3D plot indicates a sample.

It can be observed that the 16 flight states in the input layer are in a chaos. After the first three layers, increasing flight states are gradually distinguished but it is still difficult for all-states identification. From the forth layer, most of the flight states tend to be separated and the best classification effect is obtained by the last output layer. The results indicate a layer-wise improvement in classification performance by transforming the low-level signal data into high-level features.

### 5.2. Stall Detection and Alerting

The second case study applies the proposed method for pre-warning of one of the typical dangerous conditions in aerodynamics. Similarly, the parameters in CNN are optimized by GWO as shown in Table 3.

Ten trials of each identification method (the proposed method, 1D CNN, DNN, BPN) are conducted and the comparison between the four methods is shown in Figure 12 while the mean values and the standard deviations are summarized in Table 4.

It can be seen that the proposed method has the best performance in the classification of 12 dense-resolution flight states with fewer network parameters. To further examine the stall detection and alerting effects, a classification report of one the ten trials is obtained in Table 5 with three criteria: Precision, Recall, and F1 score. Precision is the ratio of correctly predicted positive observations to the total predicted positive observations, while Recall is the ratio of correctly predicted positive observations to the all observations in the actual class. F1 Score is the weighted average of Precision and Recall: F1 Score = 2 * (Recall * Precision)/(Recall + Precision) [38]. Safe, Alert, and Stall regions are divided with corresponding flight states. The overall identification accuracy (recall) is 93%.

To facilitate detailed analysis, a normalized confusion matrix is presented in Figure 13. Each row of the matrix represents the test samples in a true class label while each column indicates the samples in a predicted class label [39].

Herein, the recall value and precision value can be interpreted as the miss-alarm ratio and false alarm ratio. As shown in Figure 11, the lowest classification accuracy occurs at State 8 (Alert region), in which 19% samples belonging to State 8 are wrongly predicted as State 4 (Safe region) as can be found in the eighth row of the confusion matrix. This is denoted as the miss-alarm ratio. From another perspective, 11% samples of the predicted samples actually belong to State 4 (Safe region) as can be found in the eighth column of the confusion matrix, meaning that the wing flying in Safe region yet receives a false alarm. This partial precision value is regarded as false alarm ratio. Overall, the confusion matrix provides a detailed insight of the identification accuracy on each flight state for further improvement actions.

## 6. Conclusions

In this paper, a novel adaptive CNN is proposed for flight-state identification from structural vibration signals obtained from a self-sensing composite wing through wind tunnel experiments. A 1D deep CNN structure is established to facilitate the automatic feature extraction from the lengthy vibration signal. To improve the identification accuracy, the subsignals are decomposed and reconstructed via the DTCWPT, which is able to obtain the most differentiated subsignal components and take advantage of the multichannel mechanism of the CNN input. To avoid subjective human experience in the key parameter determination, a GWO algorithm is further attached to the CNN, which enables the self-adaptive parameter setting to extract more informative features.

The proposed method is evaluated through the general flight state identification and applied in dangerous flight states detection. Results from both cases demonstrate the superior performance of the proposed method over the conventional CNN method and general deep-learning network without modification. In all, the proposed method has the advantage of saving considerable efforts in feature extraction and parameter setting of the deep neural network by human experts and is promised to be used in broader applications for intelligent structural vibration analysis.

## Figures and Tables

**Figure 1 sensors-19-00275-f001:**
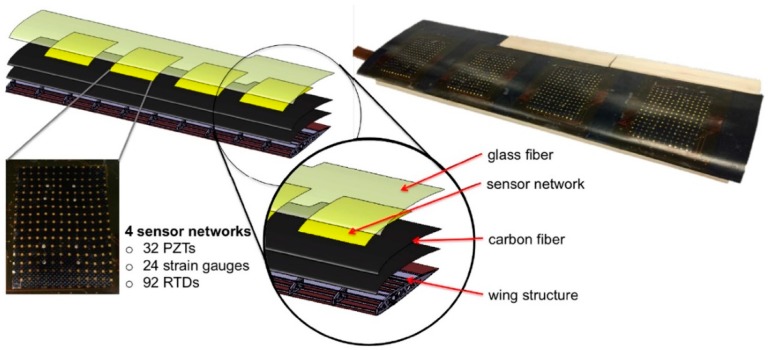
Self-sensing composite wing design [2].

**Figure 2 sensors-19-00275-f002:**
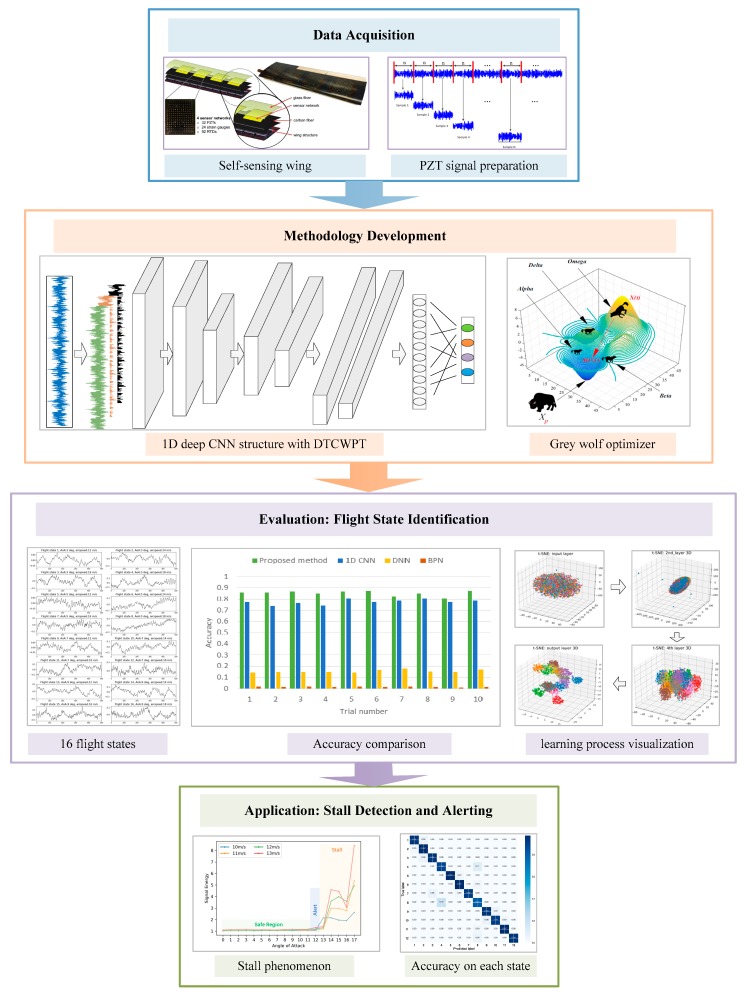
Framework of the proposed methodology.

**Figure 3 sensors-19-00275-f003:**
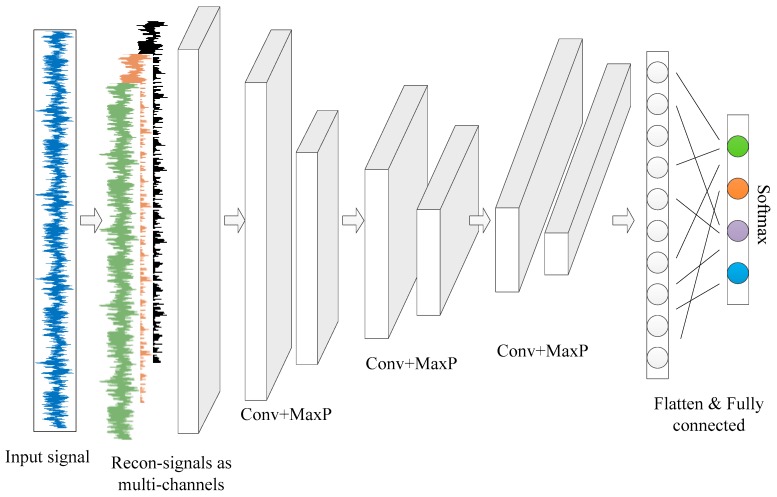
1D deep convolutional neural network (CNN) structure.

**Figure 4 sensors-19-00275-f004:**
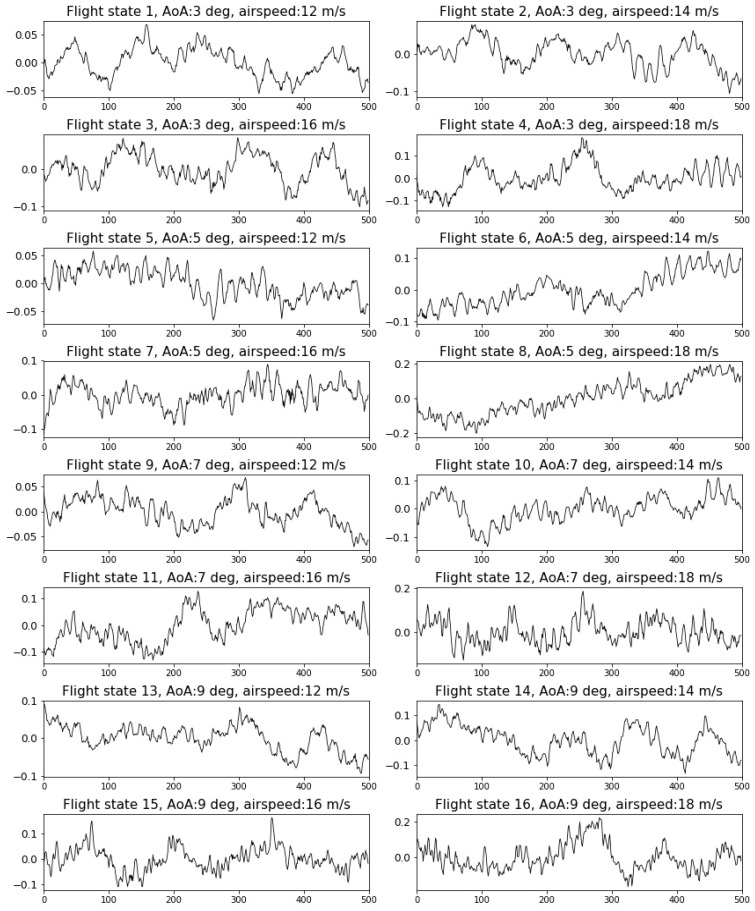
Piezoelectric lead–zirconate titanate (PZT) signal segments under 16 flight states after data preprocessing.

**Figure 5 sensors-19-00275-f005:**
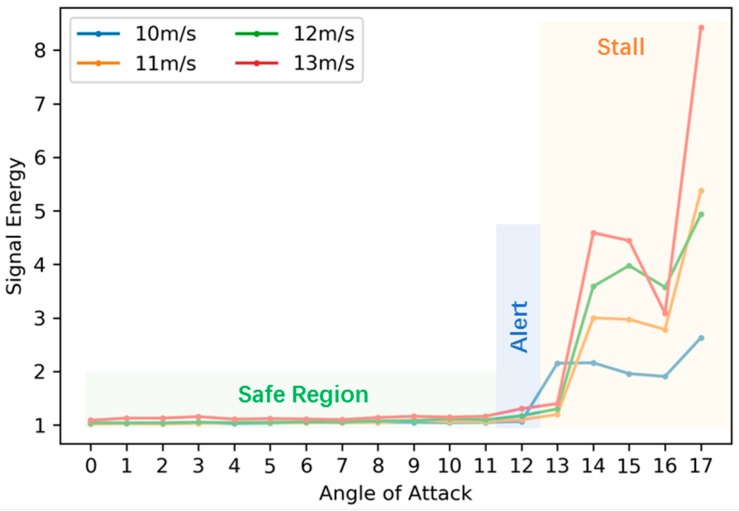
Signal energy under various flight states.

**Figure 6 sensors-19-00275-f006:**
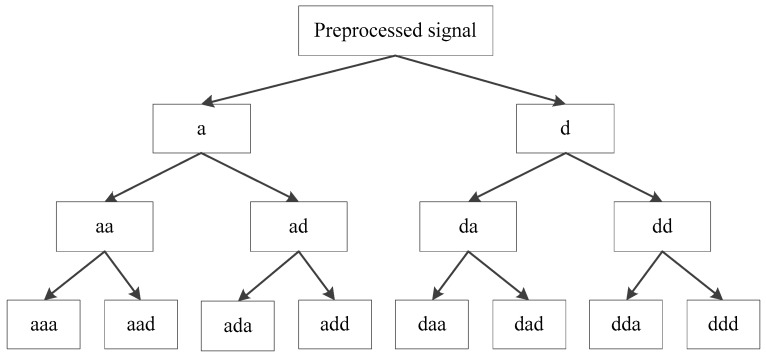
Three-level decomposition structure.

**Figure 7 sensors-19-00275-f007:**
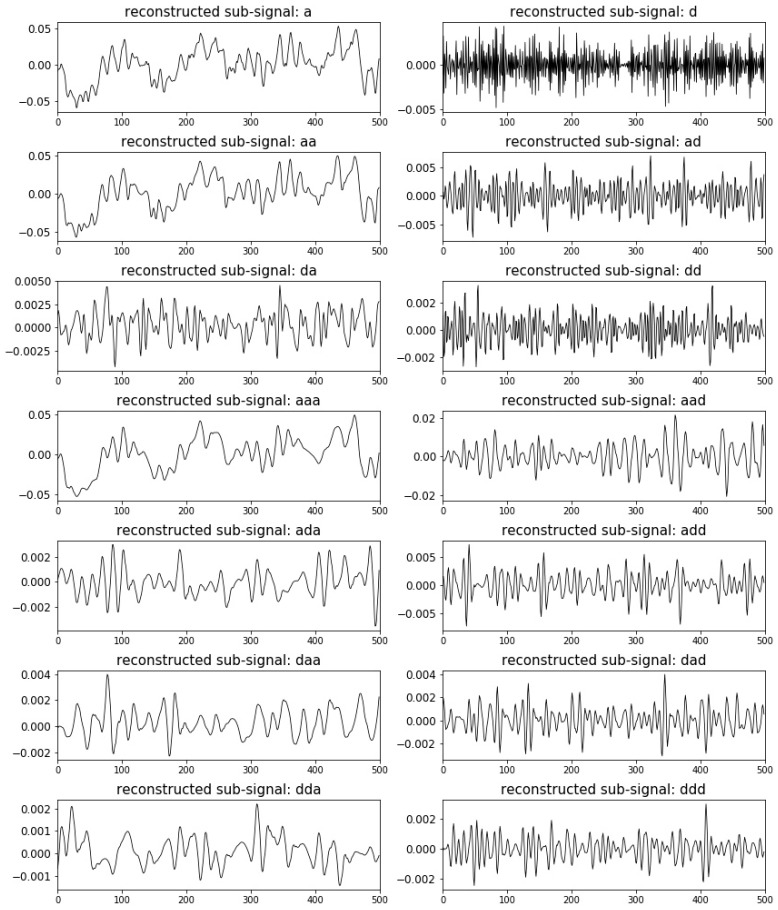
Reconstructed subsignals at different decomposition levels.

**Figure 8 sensors-19-00275-f008:**
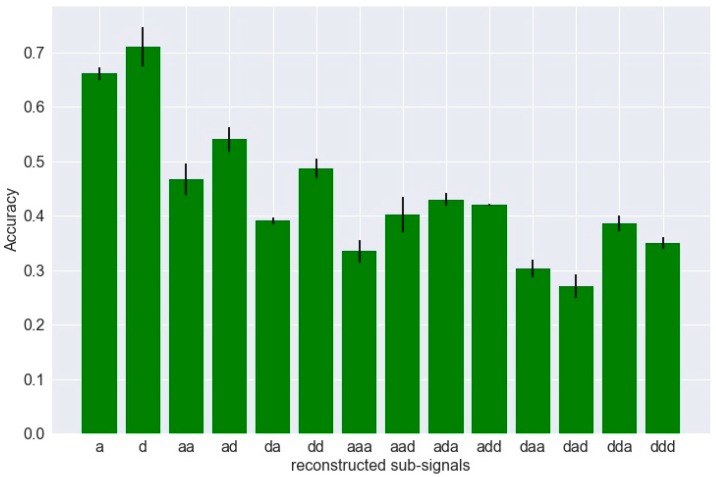
Identification accuracy under various reconstructed subsignals.

**Figure 9 sensors-19-00275-f009:**
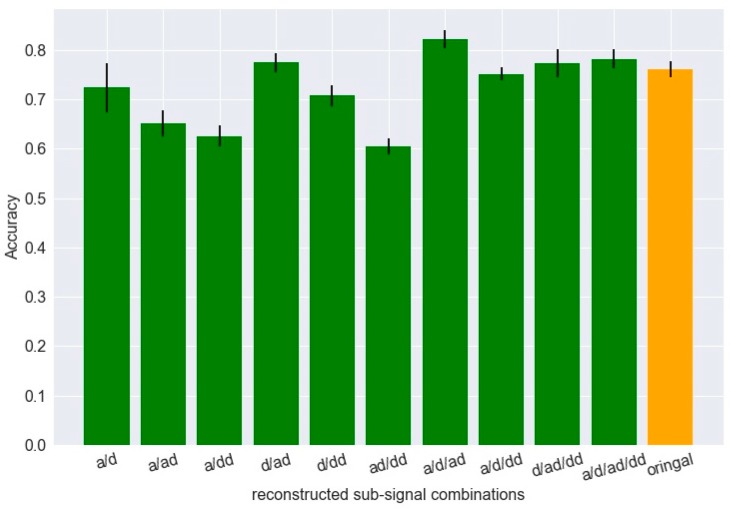
Identification accuracy under various reconstructed subsignal combinations.

**Figure 10 sensors-19-00275-f010:**
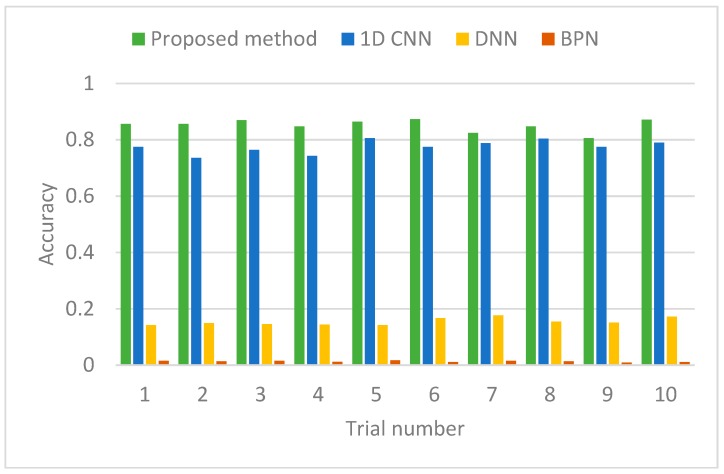
Identification accuracy of four methods.

**Figure 11 sensors-19-00275-f011:**
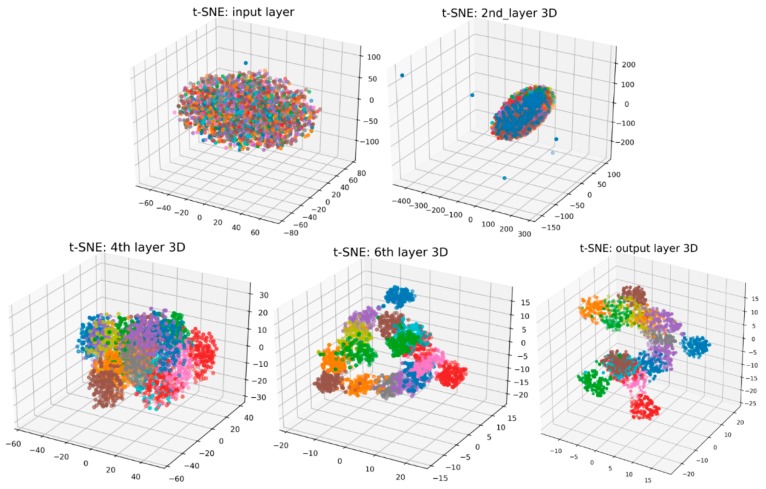
3D visualization of the hierarchical learning process.

**Figure 12 sensors-19-00275-f012:**
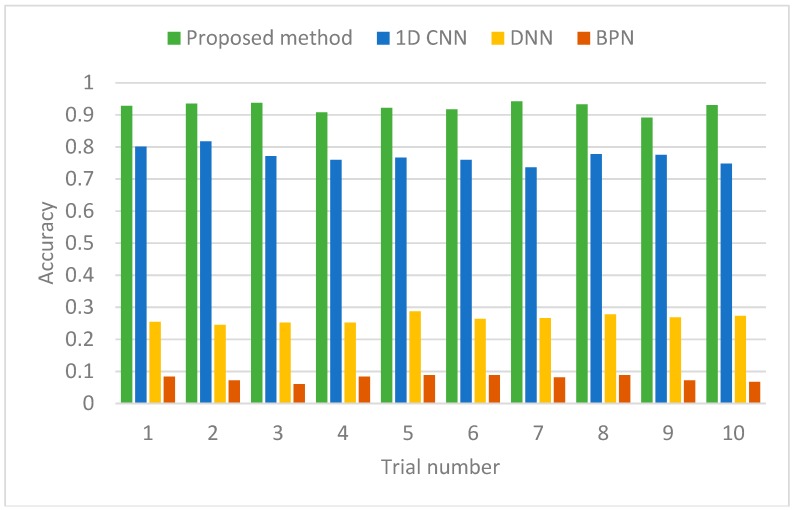
Identification accuracy of four methods.

**Figure 13 sensors-19-00275-f013:**
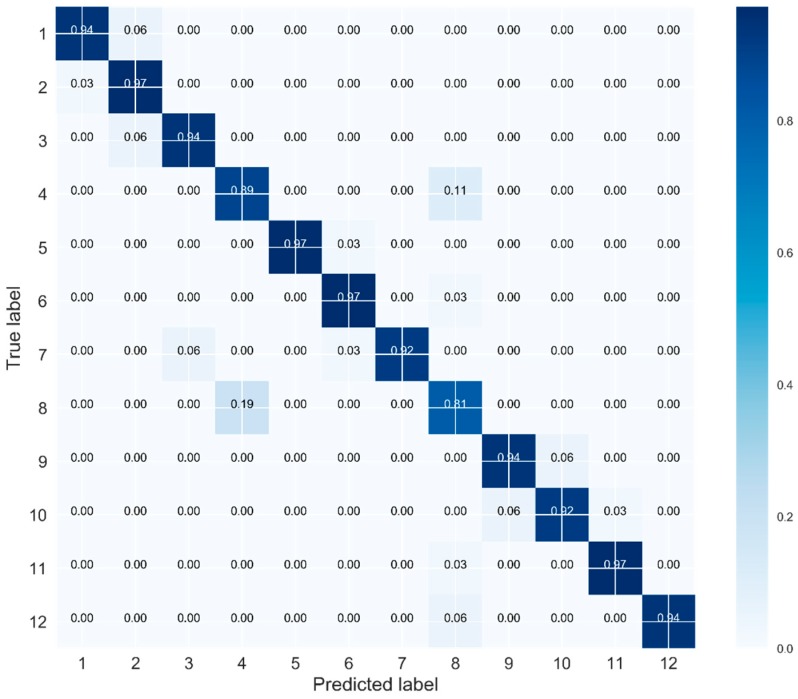
Confusion matrix of flight-state identification.

**Table 1 sensors-19-00275-t001:** Features in time domain.

Parameter Description	Value after GWO	Value Before (Used in Normal CNN in the Following Comparison)
Population of GWO	20	-
Iteration number	20	-
Dimensionality of particles	3	-
Kernel size in C2 and C3 layers	5	10
Learning rate	0.0012	0.001
Dropout rate	0.4	0.5

**Table 2 sensors-19-00275-t002:** Features in frequency domain.

Methods	Input Dimension	Size of Training/Testing Sample	Average Testing Accuracy	Standard Deviation	Total Parameters
Proposed method	500	2304/576	85.15%	2.07%	260,432
1D CNN	500	2304/576	77.53%	2.20%	500,816
DNN	500	2304/576	15.45%	1.22%	214,100
BPN	500	2304/576	1.37%	0.27%	129,000

**Table 3 sensors-19-00275-t003:** Parameters used with and without GWO.

Parameter Description	Value after GWO	Value before (Used in Normal CNN in the Following Comparison)
Population of GWO	20	-
Iteration number	20	-
Dimensionality of particles	3	-
Kernel size in C2 and C3 layers	3	10
Learning rate	0.0011	0.001
Dropout rate	0.4	0.5

**Table 4 sensors-19-00275-t004:** Average testing accuracy and the parameter numbers of four methods.

Methods	Input Dimension	Size of Training/Testing Sample	Average Testing Accuracy	Standard Deviation	Total Parameters
Proposed method	500	1728/432	92.43%	1.48%	160,588
1D CNN	500	1728/432	77.11%	2.27%	498,764
DNN	500	1728/432	26.41%	1.24%	213,700
BPN	500	1728/432	7.82%	0.94%	128,000

**Table 5 sensors-19-00275-t005:** Classification report.

	States ID	AoA deg	Speed m/s	Precision	Recall	F1 Score
Safe	1	11	10	0.97	0.94	0.96
	2	11	11	0.90	0.97	0.93
	3	11	12	0.94	0.94	0.94
	4	11	13	0.82	0.89	0.85
Alert	5	12	10	1.00	0.97	0.99
	6	12	11	0.95	0.97	0.96
	7	12	12	1.00	0.92	0.96
	8	12	13	0.78	0.81	0.79
Stall	9	13	10	0.94	0.94	0.94
	10	13	11	0.94	0.92	0.93
	11	13	12	0.97	0.97	0.97
	12	13	13	1.00	0.94	0.97

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
