# Peer review of "A Self-Adaptive 1D Convolutional Neural Network for Flight-State Identification"

_sensors, 2019, doi:10.3390/s19020275_

Round 1

Reviewer 1 Report

The article presents a 1D CNN which have been implemented by the authors. Its purpose is to extract features from the structural vibration of a brand new self-sensing wing through wind tunnel experiments. This is automatically achieved by the CNN. The study has been validated with two case studies and the results seem to be accurate and robust. The article is well structured, the topic is interesting, with a novel proposal that fits with the scope of the magazine. However, there are some issues that must be faced before acceptance for publication.

Major issues:

- Related work is not properly covered.

- The conclusion reads a bit uninspired. It is merely a summary of the paper. What are the main "Take Aways" for the reader?

Minor issues:

- If you are defining CNN in the abstract, you should also define GWO there.

- Define terms like CNN, GWO, DTCWPT or 1D just once throughout all the document.

- Figure 4 needs further description in section 4.2.

- Some numbers in the confusion matrix (Figure 13) are not very legible. Perhaps it would be better to put it in table format rather than as an image.

Author Response

Many thanks for your comments.

- Related work is not properly covered.

R: The authors stated the previous published work in the Supplementary Description. Now this work has been added in Introduction Section, which have been highlighted in yellow in page 2, line 82-84.

- The conclusion reads a bit uninspired. It is merely a summary of the paper. What are the main "Take Aways" for the reader?

R: The conclusions have been updated to highlight the benefits of the proposed method instead of merely paper summary. Details can be seen in the revised manuscript.

- If you are defining CNN in the abstract, you should also define GWO there.

R: the acronym CNN appears three times in the abstract while other words (like DTCWPT and GWO just appear once). Therefore, the authors write the whole name.

- Define terms like CNN, GWO, DTCWPT or 1D just once throughout all the document.

R: Most parts have been corrected except for Conclusion Section in case readers just browsing the abstract and conclusion.

- Figure 4 needs further description in section 4.2.

R: Further description has been added.

- Some numbers in the confusion matrix (Figure 13) are not very legible. Perhaps it would be better to put it in table format rather than as an image.

R: Since we have up to 16 classes, the numbers in the confusion matrix are a little bit dense which causes the legibility. A table of pure numbers might be difficult to observe as well. Anyway, we have enlarged the figure to some extent and the depth of the color can provide a good indication of the different accuracy value.

Reviewer 2 Report

I have reviewed the manuscript "A Self-Adaptive 1D Convolutional Neural Network for Flight State Identification", Manuscript ID: sensors-393576. In this paper, the authors propose an adaptive convolutional network (CNN) for flight state identification from structural vibration signals obtained from a self-sensing composite wing through wind tunnel experiments. The manuscript under review is interesting, documented and generally well structured. However, I consider that the article will benefit if the authors take into account the following remarks and address within the manuscript the signaled issues:

1.     In the Supplementary Description that the authors have provided on the MDPI upload form, they have stated: "This submitted manuscript shares the same background and experiment data with the previous published one. The difference is the methodology development, results and discussion." Consequently, the authors have succeeded in establishing a precedence in their line of research on the topic of Flight State Identification. Therefore, I consider that the manuscript under review will benefit a lot if the authors cite in the discussion section their previous paper and highlight how their research has evolved from the findings of their previous study from the Journal Sensors, namely "Chen, X.; Kopsaftopoulos, F.; Wu, Q.; Ren, H.; Chang, F.-K. Flight State Identification of a Self-Sensing Wing via an Improved Feature Selection Method and Machine Learning Approaches. Sensors 2018, 18, 1379.", to the present research results reported in the current manuscript.

2.     Lines 33-128: Even if the authors have performed a critical survey of what has been done up to this point in the scientific literature in the "Introduction" section, they did not emphasize enough a clear gap in the current state of knowledge that needs to be filled, a gap that is being addressed by their manuscript. 

3.     Lines 120-123: The authors have presented the framework of their proposed methodology in the Introduction section (Figure 2). I consider that this figure is more suitable for the "Materials and Methods" section. Moreover, in the actual form of the paper the "Materials and Methods" section title is missing, but its content is contained by the sections "2. Problem Statement" and "3. Methodology Development" (Lines 129-267). I consider that these two sections must be concatenated and structured into the "Materials and Methods" section, as requested by the Sensors MDPI Journal's Template.

4.     Lines 174, 188, 194, 199, 241, 242, 248-251:  The equations within the manuscript should be explained, demonstrated or cited, as there are some equations that have not been introduced in the literature for the first time by the authors and that are not cited.

5.     Line 160: It will benefit the paper to specify, in the new "Materials and Methods" section, details regarding the version numbers for the software and the detailed hardware configuration used within the research.

6.     Lines 268-318: As the section "Case study" contains two case studies including the general flight state identification and the stall detection and alerting, I consider that this information is more appropriate for the "Results and Discussion" section and therefore the section "Case study" and the "Results and Discussion" one must be concatenated into a single one, under the name "Results and Discussion", as requested by the Sensors MDPI Journal's Template.

7.     Lines 54, 121, 214, 298, 308, 325, 327, 338, 347, 350, 372, 392, 411, 428, 434: According to the Sensors MDPI Journal's Template, all the figures should be cited in the main text as Figure 1, Figure 2, etc. In the manuscript under review, this information appears in the main text as "Fig. …". Please address this issue by modifying the way in which the figures are referred in the main text, according to the Energies MDPI Journal's Template. Line 330: At Figure 7, the titles and the measurement units of both axes are missing in all the cases.

8.     Line 441: The paper will benefit if the authors make a step further, beyond their approach and provide an insight at the end of the "Results and Discussion" section regarding what they consider to be, based on the obtained results, the most important steps that all the involved parties should take in order to benefit from the results of the research conducted within the manuscript.

9.     Line 472-561: Regarding the format of the paper: the text is not formatted using the Justify alignment in the References section. Please address these issues, according to the Sensors MDPI Journal's Template.

10.  Line 37: "temperature, pressure, aerodynamic forces, etc.", Line 44: "…flutter, stall, etc.", Lines 65-66: "…altitude, angle of attack, airspeed and temperature, etc.", Line 81: "…the mean value, standard deviation, peak value, kurtosis, etc.", Lines 90-91: "…deep belief network (DBN) [23], convolutional neural network (CNN) [24], recurrent neural network (RNN) [25], etc.", Line 167: "…computer vision [34], speech recognition [35], and fault diagnosis [24], etc.", Lines 235-236: " well-known evolutionary algorithms such as particle swarm optimization, genetic algorithm, differential evolution, etc." In a scientific paper one should avoid using run-on expressions, such as "and so forth", "and so on" or "etc.". Therefore, instead of "etc.", the sentences should mention all the characteristics that have been taken into account in the study, as they are relevant to the analysis and to the obtained results. 

Author Response

Many thanks for your detailed comments.

1.     In the Supplementary Description that the authors have provided on the MDPI upload form, they have stated: "This submitted manuscript shares the same background and experiment data with the previous published one. The difference is the methodology development, results and discussion." Consequently, the authors have succeeded in establishing a precedence in their line of research on the topic of Flight State Identification. Therefore, I consider that the manuscript under review will benefit a lot if the authors cite in the discussion section their previous paper and highlight how their research has evolved from the findings of their previous study from the Journal Sensors, namely "Chen, X.; Kopsaftopoulos, F.; Wu, Q.; Ren, H.; Chang, F.-K. Flight State Identification of a Self-Sensing Wing via an Improved Feature Selection Method and Machine Learning Approaches. Sensors 2018, 18, 1379.", to the present research results reported in the current manuscript.

R: the previous work has been cited in the introduction section indicating the difference between two studies. The reason why we did not put it in the discussion section is that the results from two studies are not comparable. The previous study addressed the feature extraction and selection methods which heavily rely on experts’ knowledge while the methodology developed in this study intends to get rid of the human intervene and try to be self-adaptive. Therefore, we highlight the difference in the introduction section.

2.     Lines 33-128: Even if the authors have performed a critical survey of what has been done up to this point in the scientific literature in the "Introduction" section, they did not emphasize enough a clear gap in the current state of knowledge that needs to be filled, a gap that is being addressed by their manuscript. 

R: Although this paper addresses a class of signal: vibration signal, instead of common mechanics such as bearing and gears, the vibration source this study is quite novel, which is from a self-sensing composite wing developed at the laboratory at Stanford University. The vibration includes some complicated coupled structural aerodynamic behavior of the self-sensing wing. Therefore, instead of a clear gap in the current state of knowledge that needs to be filled, there are many interest research topics (e.g. flight state identification, wing vibration prediction, etc.) to address since the overall experiment is brand new. This study addresses one of the many topics the flight state identification by developing an improved deep learning method.

3.     Lines 120-123: The authors have presented the framework of their proposed methodology in the Introduction section (Figure 2). I consider that this figure is more suitable for the "Materials and Methods" section. Moreover, in the actual form of the paper the "Materials and Methods" section title is missing, but its content is contained by the sections "2. Problem Statement" and "3. Methodology Development" (Lines 129-267). I consider that these two sections must be concatenated and structured into the "Materials and Methods" section, as requested by the Sensors MDPI Journal's Template.

6.     Lines 268-318: As the section "Case study" contains two case studies including the general flight state identification and the stall detection and alerting, I consider that this information is more appropriate for the "Results and Discussion" section and therefore the section "Case study" and the "Results and Discussion" one must be concatenated into a single one, under the name "Results and Discussion", as requested by the Sensors MDPI Journal's Template.

R: The configuration of the manuscript is advised and modified by the last two authors who are our project supervisors and revised the manuscript before the first submission. They insist adding the Problem Statement Section to clearly identify the research problem. In terms of the ‘Materials and Methods’, since this study focuses on the deep learning algorithm development instead of the self-sensing wing and wind tunnel experiments, we think it would be better to use ‘Methodology Development’. As to ‘Case Study’ and ‘Results and Discussion’, we found that if we put them together, the ‘Results and Discussion’ would be too lengthy and we still need to describe the two cases before reaching the final results, therefore, we created ‘Case Study’ to make it clearer and more readable.   

4.     Lines 174, 188, 194, 199, 241, 242, 248-251:  The equations within the manuscript should be explained, demonstrated or cited, as there are some equations that have not been introduced in the literature for the first time by the authors and that are not cited.

R: The equations in the Section 3.1 mainly cover the basic theory of the CNN which can be found in many literature. We have added one citation at the end of the first equation. The rest of equations are mainly on GWO, instead of citing each equation, we have cited the source before all equations on GWO.

5.     Line 160: It will benefit the paper to specify, in the new "Materials and Methods" section, details regarding the version numbers for the software and the detailed hardware configuration used within the research.

R: the software we used in this study is Python 3.x, any version with tensorflow environment would be OK. We have specified it in the manuscript.

7.     Lines 54, 121, 214, 298, 308, 325, 327, 338, 347, 350, 372, 392, 411, 428, 434:According to the Sensors MDPI Journal's Template, all the figures should be cited in the main text as Figure 1, Figure 2, etc. In the manuscript under review, this information appears in the main text as "Fig. …". Please address this issue by modifying the way in which the figures are referred in the main text, according to the Energies MDPI Journal's Template. Line 330: At Figure 7, the titles and the measurement units of both axes are missing in all the cases.

R: ‘Fig’ problem has been fixed in the manuscript. Also, more information has been provided to Figure 7 (not on the subplots since each plot is small)

8.     Line 441: The paper will benefit if the authors make a step further, beyond their approach and provide an insight at the end of the "Results and Discussion" section regarding what they consider to be, based on the obtained results, the most important steps that all the involved parties should take in order to benefit from the results of the research conducted within the manuscript

R: Thanks for this comment, we have modified the Conclusion Section and conclude the benefits we proposed such method involving different parties.

9.     Line 472-561: Regarding the format of the paper: the text is not formatted using the Justify alignment in the References section. Please address these issues, according to the Sensors MDPI Journal's Template.

R: The Reference Section has been updated using the MDPI format by Mendeley.

10.  Line 37: "temperature, pressure, aerodynamic forces, etc.", Line 44: "…flutter, stall, etc.", Lines 65-66: "…altitude, angle of attack, airspeed and temperature, etc.", Line 81:"…the mean value, standard deviation, peak value, kurtosis, etc.", Lines 90-91: "…deep belief network (DBN) [23], convolutional neural network (CNN) [24], recurrent neural network (RNN) [25], etc.", Line 167: "…computer vision [34], speech recognition [35], and fault diagnosis [24], etc.", Lines 235-236: " well-known evolutionary algorithms such as particle swarm optimization, genetic algorithm, differential evolution, etc." In a scientific paper one should avoid using run-on expressions, such as "and so forth", "and so on" or "etc.". Therefore, instead of "etc.", the sentences should mention all the characteristics that have been taken into account in the study, as they are relevant to the analysis and to the obtained results.

R: Thanks for the advice. We have modified them in the revised manuscript.
